# Iron Status, Anemia, and Functional Capacity in Adults with Congenital Heart Disease

**DOI:** 10.3390/diagnostics15131672

**Published:** 2025-06-30

**Authors:** Raphael Phinicarides, Isabelle Esther Reuter, Georg Wolff, Athanasios Karathanos, Houtan Heidari, Maryna Masyuk, Frank Pillekamp, Malte Kelm, Tobias Zeus, Kathrin Klein

**Affiliations:** 1University Hospital Düsseldorf, Medical Faculty, Division of Cardiology, Department of Cardiology, Pulmonology and Vascular Medicine, Heinrich-Heine University, Moorenstr. 5, 40225 Düsseldorf, Germany; isabelle.simon@uni-duesseldorf.de (I.E.R.); guwolff@hotmail.com (G.W.); houtan.heidari@med.uni-duesseldorf.de (H.H.); maryna.masyuk@med.uni-duesseldorf.de (M.M.); malte.kelm@med.uni-duesseldorf.de (M.K.); kathrin.klein@med.uni-duesseldorf.de (K.K.); 2Klinik für Kardiologie, Klinikum Ibbenbüren, 49477 Ibbenbüren, Germany; 3Department of Sports Medicine, Charité Universitätsmedizin Berlin, 10115 Berlin, Germany; ath.karathanos@gmail.com; 4Department of Diagnostic and Interventional Radiology, Medical Faculty and University Hospital Düsseldorf, Heinrich-Heine-University Düsseldorf, 40225 Düsseldorf, Germany; frank.pillekamp@med.uni-duesseldorf.de; 5CARID (Cardiovascular Research Institute Düsseldorf), 40225 Düsseldorf, Germany

**Keywords:** ACHD, iron deficiency, anemia, VO_2_ max, functional capacity

## Abstract

**Background:** Congenital heart disease (CHD) affects approximately 9 per 1000 live births worldwide, with increasing prevalence due to improved survival. Today, over 90% of individuals with CHD reach adulthood, resulting in a growing population of adults with congenital heart disease (ACHD). Despite its clinical relevance, iron deficiency (ID) and anemia have been insufficiently studied in this group. **Objectives:** To evaluate the prevalence and clinical impact of iron deficiency and anemia in ACHD, particularly their relationship with exercise capacity. **Methods:** We retrospectively analyzed 310 ACHD patients at University Hospital Düsseldorf between January 2017 and January 2019. Iron status was assessed using serum ferritin, transferrin saturation (TSAT), and hemoglobin levels. Exercise capacity was measured by cardiopulmonary exercise testing (VO_2_ max). Prevalence and clinical associations were compared with those reported in heart failure populations, using ESC guideline criteria. Analyses were adjusted for age, sex, and defect complexity. **Results:** Iron deficiency (ID) was present in 183 patients (59.0%). Anemia was observed in 13 patients (4.2%), with 6 (46.2%) classified as microcytic and 5 (38.5%) as normocytic. Reduced exercise capacity, defined as VO_2_ max <80% of predicted, was present in 51 patients (16.5%), occurring more frequently in those with complex CHD (31.3% vs. 11.3%, *p* < 0.001). ID was associated with a trend toward lower VO_2_ max (21.3 vs. 23.5 mL/min/kg, *p* = 0.068), while anemia correlated with significantly reduced performance (19.8 ± 4.1 vs. 22.9 ± 6.3 mL/min/kg, *p* = 0.041). **Conclusions:** Iron deficiency is highly prevalent, and anemia—though less common—was consistently associated with reduced functional capacity in ACHD. These findings highlight the need for targeted screening and management strategies in this growing patient population.

## 1. Introduction

Congenital heart disease (CHD) affects approximately 9 per 1000 live births globally, with prevalence increasing due to improved survival into adulthood [1,2,3]. Advancements in surgical techniques, medical therapy, and interventional cardiology have enabled more than 90% of CHD patients to survive into adulthood, resulting in a growing population of adults with congenital heart disease (ACHD) that now outnumbers the pediatric CHD population [4,5].

Despite these gains, ACHD patients remain vulnerable to long-term complications, including iron deficiency (ID) and anemia. While ID in chronic heart failure (HF) is a well-established contributor to reduced exercise capacity, increased symptom burden, and adverse outcomes, with current guidelines recommending intravenous iron replacement, the prevalence and clinical implications of ID and anemia in ACHD remain insufficiently defined [6,7,8]. Pathophysiologic mechanisms such as cyanotic physiology, chronic inflammation, erythropoietic dysregulation, and renal dysfunction may predispose ACHD patients to hematologic abnormalities that impair functional capacity and quality of life.

Evidence on the prevalence, phenotype, and clinical relevance of ID and anemia in ACHD is limited. Small-scale studies have suggested possible associations with impaired oxygen uptake and adverse outcomes, particularly in cyanotic individuals, but results have been inconsistent.

This study aimed to assess the prevalence of iron deficiency and anemia in a well-characterized ACHD cohort and to examine their association with objective measures of functional capacity. Our goal was to determine whether these potentially modifiable hematologic conditions contribute to impaired exercise performance. The results could guide future strategies for risk stratification and therapeutic management.

## 2. Materials and Methods

This study is a retrospective analysis of patients followed at the certified Adult Congenital Heart Disease (ACHD) outpatient clinic of the University Hospital Düsseldorf (UKD), accredited by the German Society of Cardiology. All patients attending the clinic between January 2017 and January 2019 were included, provided that complete documentation of their visit and relevant iron metabolism parameters was available.

The study received ethics approval from the Ethics Committee of Heinrich Heine University Düsseldorf (Reference 2019-375-RetroDEuA; approved 30 January 2019). All procedures conformed to ethical standards for retrospective analyses. No animal experiments were conducted or required.

To be eligible, patients had to have a confirmed diagnosis of congenital heart disease, regardless of surgical status. Inclusion required comprehensive clinical documentation, including blood counts and iron status parameters. Patients were excluded if iron status parameters (ferritin and transferrin saturation) were not available. This applied to 14 patient visits, which were removed prior to deduplication to avoid retaining incomplete records.

The extracted baseline variables included date of visit, age, sex, height, weight, and body mass index (BMI). Each patient was assigned a unique identifier to avoid duplication. Data were analyzed anonymously.

CHD lesions were classified per ESC guidelines into four categories: shunt lesions, left-sided obstruction, right-sided obstruction, and complex malformations. In cases with multiple anomalies, the lesion with the greatest hemodynamic impact was designated as the primary diagnosis. Patients with Eisenmenger physiology were classified as complex, regardless of origin. The systemic ventricle and presence of univentricular physiology were also recorded.

Documentation included prior surgical repair, cyanotic status, and Eisenmenger syndrome. Cyanosis was defined as SpO_2_ ≤ 92% on room air or documented clinical hypoxemia.

Systolic ventricular function was categorized based on ejection fraction (EF): normal (≥55%), mildly reduced (45–54%), moderately reduced (30–44%), and severely reduced (<30%) [9]. Valve function was graded (mild/moderate/severe stenosis or regurgitation) per ESC echocardiographic criteria using Doppler-based parameters. MRI findings were prioritized for ventricular function; echocardiography was used primarily for valvular assessment (GE Healthcare, Chicago, IL, USA).

The New York Heart Association (NYHA) functional class was assigned based on clinical records or estimated from reported exercise tolerance [9].

Additional symptoms—including dyspnea, fatigue, chest pain, lightheadedness, and headache—were extracted. Physical capacity was assessed via cardiopulmonary exercise testing (CPET), bicycle ergometry, or a six-minute walk test (6MWT) based on clinical appropriateness [10]. CPET included VO_2_ max using a breath-by-breath gas exchange on a cycle ergometer (Vyaire Medical GmbH, Hoechberg, Germany). Ergometry data included maximal metabolic equivalents (METs) (Ergoline GmbH, Bitz, Germany). The 6MWT was performed on a standardized 30-m indoor track, recording the total distance walked. Heart rate, SpO_2_, and exertion ratings were documented when available (Nonin Medical, Inc., Plymouth, MN, USA).

Blood samples were routinely drawn at each visit, and laboratory results were recorded (Roche Diagnostics GmbH, Mannheim, Germany). In the first phase, patients were stratified into cyanotic and acyanotic groups, with comparative analyses focusing on iron deficiency, anemia, and functional capacity. The goal was to explore phenotypic differences and pathophysiological patterns between these subgroups.

In the second phase, patients were divided by anemia status, and the relationship with iron parameters and cyanotic/acyanotic phenotype was analyzed. This integrative approach aimed to explore how anemia and CHD morphology jointly influence functional outcomes.

Chronic kidney disease (CKD) was defined as eGFR < 60 mL/min/1.73 m^2^ on two occasions or per documented diagnosis.

Anemia was defined according to WHO and ESC guidelines [9,11,12]: Hb < 12 g/dL in women and <13 g/dL in men. Mean corpuscular volume (MCV) categorizes anemia as microcytic (≤80 fL), normocytic (80–96 fL), or macrocytic (≥96 fL).

Iron deficiency (ID) was defined as [9,13]
Serum ferritin < 100 µg/L (absolute ID);Serum ferritin 100–299 µg/L with TSAT < 20% (functional ID).

The soluble transferrin receptor (sTfR)/log ferritin ratio was additionally used where available.

Patients with both anemia and ID were classified as having iron deficiency anemia (IDA).

While these criteria are well established in chronic heart failure populations [9,14], their applicability in ACHD remains uncertain. In cyanotic patients or those with erythrocytosis (e.g., Eisenmenger syndrome), conventional markers such as ferritin and TSAT may be confounded by inflammation, hypoxia, or altered iron metabolism [1]. These diagnostic thresholds have not yet been validated in ACHD cohorts and should therefore be interpreted with caution [7,10,15,16,17,18].

Comorbidities and medications were extracted from records. If not explicitly listed, comorbidities were assumed to be absent. 


**Statistical analysis:**


The collected dataset was analyzed using IBM SPSS Statistics, Version 26 (IBM Corp., Armonk, NY, USA), and included metric, ordinal, and nominal variables.

Metric variables were summarized as mean ± standard deviation (SD), while categorical variables were reported as absolute and relative frequencies.

To compare two groups, a series of statistical tests was applied based on data distribution and scale. Metric variables were tested for normal or log-normal distribution using the Shapiro–Wilk test. Depending on the distribution, either independent samples t-tests (for normally distributed data) or Mann–Whitney U tests (for non-normally distributed data) were used for comparisons between two groups. For analyses involving more than two groups, one-way analysis of variance (ANOVA) was applied if the assumptions for parametric testing were met. Associations between categorical variables were assessed using the chi-square test or Fisher’s exact test when the expected cell frequencies were ≤5. Correlations between variables were evaluated using Pearson’s correlation for normally distributed metric variables and Spearman’s rank correlation in cases of ordinal data or when at least one metric variable was not normally distributed. All statistical tests were two-tailed, and a *p*-value of <0.05 was considered statistically significant.

## 3. Results

Between January 2017 and January 2019, a total of 525 patient visits were screened. Fourteen visits were excluded due to missing iron panel data (ferritin and transferrin saturation). Among the remaining visits, 215 were identified as repeat follow-up consultations and excluded to ensure each patient was represented only once. The final study cohort consisted of 310 unique ACHD patients with complete clinical, laboratory, and functional data.

The cohort comprised 160 male (51.6%) and 150 female (48.4%) patients, with an age range of 17 to 77 years (mean age: 33 ± 13.3 years). The mean body mass index (BMI) was 25.1 ± 4.7 kg/m^2^.

All major congenital heart defects listed by the Competence Network for Congenital Heart Disease were represented in the cohort (see Table 1). Additionally, less common defects were documented, including pulmonary vein anomalies, aortopathies, congenital mitral valve malformations, double-chambered right ventricles, Ebstein anomaly, and coronary artery anomalies. Due to the presence of multiple cardiac anomalies in some patients, the cumulative frequency of individual diagnoses may exceed 100%.

Patients were distributed across the four principal CHD categories defined by ESC guidelines: shunt lesions, left-sided malformations, right-sided malformations, and complex malformations, with sex distribution across categories presented in Table 2.

Among the study population, 202 patients (65.2%) had primarily acyanotic heart defects, and 108 patients (34.8%) were classified as cyanotic. Comorbid conditions and their distribution across CHD subtypes are summarized in Table 3.

Baseline characteristics, including age, height, weight, and BMI, are summarized as means ± standard deviation in Table 1. As not all patients had complete anthropometric data, the sample size (N) varied accordingly. BMI was calculated as weight in kilograms divided by height in meters squared (kg/m^2^).

In cases of historical cyanosis with subsequent surgical repair or when documentation was incomplete or ambiguous, patients were categorized as ‘not classified’ (Figure 1). Each patient was assigned to only one phenotype category to ensure non-overlapping group comparisons.

The cyanotic vs. acyanotic classification was determined using a combination of resting SpO_2_, anatomical and physiological features, and clinical documentation. In cases of uncertainty—such as patients with normalized oxygen saturation following repair or insufficient historical records—classification was deferred to the ‘not classified’ category.

Anemia was identified in 13 patients (4.2%), including 4 male (2.5%) and 9 female (6.0%) patients (Figure 2). Among these, 6 had microcytic and 7 had normocytic anemia. Although the number of anemic patients in our cohort was small (*n* = 13), this aligns with prior ACHD studies, including Dimopoulos et al. (4.2%) and Rodríguez-Hernández et al. (6%) [19,20]. Despite the limited sample size, significant associations with exercise capacity, NT-proBNP, and symptom burden were still detected, underlining the clinical relevance of even mild anemia in ACHD.

Anemic patients exhibited significantly impaired exercise tolerance, as reflected by increased dyspnea (*p* = 0.019), higher frequency of NYHA class > I (*p* = 0.045), and elevated NT-proBNP levels (651.2 ± 831.2 ng/L vs. 179.8 ± 353.0 ng/L, *p* = 0.010) compared to non-anemic individuals.

No significant associations were observed between anemia and age, sex, weight, CHD subtype, or systemic ventricular ejection fraction (EF). However, graphical analysis of spiroergometry data revealed a trend toward lower peak VO_2_ in anemic patients, with one outlier demonstrating preserved exercise capacity. When excluding four patients with Eisenmenger syndrome, who had secondary erythrocytosis and were not classified as anemic, the association between anemia and peak VO_2_ became statistically significant (*p* = 0.041).

Anemia was also significantly associated with the use of ACE inhibitors (*p* = 0.042), AT1 receptor antagonists (*p* = 0.012), beta-blockers (*p* = 0.023), and diuretics (*p* = 0.006). Additionally, chronic kidney disease was significantly more frequent in anemic patients (*p* = 0.022). The overall medication use profile of the cohort is detailed in Table 4.

The corresponding laboratory parameters, including hemoglobin levels, ferritin, transferrin saturation, and iron status, are summarized in Table 5.

While no association with exercise metrics such as max VO_2_ (*p* = 0.058) or METs in ergometry (*p* = 0.110) was initially found, a statistically significant relationship between anemia and max VO_2_ emerged after the exclusion of Eisenmenger patients.

Patients with Eisenmenger syndrome and secondary erythrocytosis demonstrated poor functional performance, contributing to the distortion of the correlation (Figure 3). After excluding these four patients, the association between Hb concentration and peak VO_2_ became even stronger (r = 0.309, *p* < 0.001).

Patients with anemia achieved only 71 ± 12% of the predicted VO_2_ max, compared to 89 ± 15% in non-anemic patients (*p* = 0.041, after excluding Eisenmenger patients).

In total, 259 patients (83.5%) were classified as NYHA Class I, reporting no limitations in physical activity. Overall, 51 patients (16.5%) reported exercise intolerance (NYHA II–IV). The highest prevalence of NYHA class > I was observed in patients with complex congenital heart defects (26%), followed by shunt lesions (18%), right-sided defects (17%), and left-sided lesions (5%).

Dyspnea was reported in 41 patients (13.2%), dizziness in 20 (6.5%), chest pain in 13 (4.2%), headaches in 9 (2.9%), and fatigue in 8 (2.6%).

Regarding systemic ventricular function, most patients had a normal ejection fraction (EF ≥ 55%). In 110 patients with available EF data, the mean EF was 59 ± 7% (range: 32–80%; median: 60%). Approximately 15% of the cohort had some degree of systolic dysfunction.

Sex-based differences were evident in exercise testing: males outperformed females in both spiroergometry (*p* < 0.001) and ergometry (*p* = 0.004).

In 154 patients undergoing CPET, VO_2_ max ranged from 7.8 to 38.6 mL/min/kg (mean: 22.7 ± 6.4 mL/min/kg; median: 22.3 mL/min/kg). Patients with NYHA II–IV had significantly lower VO_2_ max (16.7 ± 4.5 vs. 24.2 ± 6.0 mL/min/kg, *p* < 0.001) (Figure 4).

In 97 patients who performed ergometry, METs ranged from 3.1 to 13.0 (mean: 7.7 ± 2.2, median: 7.8). NYHA II–IV patients achieved only 4.5 ± 0.6 METs, compared to 8.0 ± 2.0 METs in NYHA I patients (*p* < 0.001) (Figure 5).

Seventeen patients completed a six-minute walk test (6MWT). Distances ranged from 123 to 538 m (mean: 366 ± 113 m, median: 395 m). NYHA II–IV patients walked 359 m, while NYHA I patients walked 378 m—this difference was not statistically significant (*p* = 0.746). Overall, 7 of the 17 patients undergoing 6MWT had Eisenmenger syndrome.

Iron Deficiency (ID) and Associated Characteristics

Using the definition from Peyrin-Biroulet et al. [13] (TSAT < 20%), patients with ID had lower body weight (70.5 ± 17.6 kg vs. 75.0 ± 16.8 kg, *p* = 0.011), were more often female (*p* < 0.001), and were younger (*p* = 0.003). They also more frequently reported headaches (*p* = 0.021).

ID was associated with lower hemoglobin levels (14.1 ± 2.0 vs. 15.1 ± 1.8 g/dL, *p* < 0.001), a higher prevalence of anemia (*p* = 0.014), and reduced mean corpuscular volume (86.3 ± 5.1 vs. 88.5 ± 5.1 fL, *p* < 0.001).

There were no significant associations between ID and METs (*p* = 0.364), 6MWT distance (*p* = 0.352), NYHA class (*p* = 0.544), NT-proBNP (*p* = 0.859), EF (*p* = 0.383), or clinical symptoms including angina (*p* = 0.762), dyspnea (*p* = 0.447), dizziness (*p* = 0.948), or fatigue (*p* = 0.697).

Eisenmenger syndrome was not significantly associated with ID (*p* = 0.725). No associations were observed between ID and ferritin (*p* = 0.329) or TSAT (*p* = 0.366) in Eisenmenger patients. However, these patients had significantly higher soluble transferrin receptor (sTfR) levels (*p* < 0.001).

Ferritin levels did not correlate with max. VO_2_ (*p* = 0.572) or the 6MWT distance (*p* = 0.440) but did correlate positively with METs in ergometry (r = 0.201, *p* = 0.048).

Anemia, Exercise Capacity, and Medication Use

Among the 13 anemic patients, trends toward increased dyspnea, elevated NT-proBNP, and higher NYHA class were noted, although these findings should be interpreted cautiously due to the small sample size.

Anemic patients had significantly lower MCV (78.3 ± 11.2 vs. 88.2 ± 4.3 fL, *p* = 0.001), lower TSAT (16.2 ± 13.7% vs. 27.0 ± 11.8%, *p* = 0.003), and lower ferritin (102.7 ± 764.3 µg/L vs. 157.8 ± 154.7 µg/L, *p* = 0.030).

Anemia was significantly associated with chronic kidney disease (*p* = 0.022) and with the use of several cardiovascular medications, as follows:ACE inhibitors (*p* = 0.042);AT1 receptor antagonists (*p* = 0.012);Beta-blockers (*p* = 0.023);Diuretics (*p* = 0.006).

No significant associations were observed between anemia and age (*p* = 0.331), sex (*p* = 0.124), weight (*p* = 0.887), CHD lesion group (*p* = 0.428), or systemic ventricular EF (*p* = 0.864).

Apart from dyspnea, no other symptoms were significantly associated with anemia. No significant differences were seen in VO_2_ max (*p* = 0.058) or METs (*p* = 0.110), although graphical comparison showed that anemic patients tended to perform worse. One outlier with excellent capacity was noted.

Patients with secondary erythrocytosis from Eisenmenger syndrome had poor functional status but were not classified as anemic. Excluding these individuals yielded a significant correlation between anemia and max. VO_2_ (*p* = 0.041). Similarly, lower hemoglobin levels correlated significantly with reduced exercise capacity, both for max. VO_2_ (r = 0.221, *p* = 0.006) and METs (r = 0.292, *p* = 0.004).

## 4. Discussion

This retrospective cohort study investigated the association between iron deficiency (ID), anemia, and exercise capacity in adults with congenital heart disease (ACHD). The primary objectives were to assess the prevalence of ID and anemia in this population and to evaluate their impact on physical performance.

The three key findings of this study are as follows:Iron deficiency and anemia were prevalent in this ACHD cohort, affecting 59.0% (183/310) and 4.2% (13/310) of patients, respectively. A significant association was observed between iron deficiency and lower hemoglobin levels (13.7 ± 1.5 g/dL vs. 14.4 ± 1.4 g/dL in non-ID, *p* < 0.001), indicating a continuum from iron depletion to overt anemia;Anemia in ACHD patients was significantly associated with reduced exercise capacity (peak VO_2_: 19.8 ± 4.1 vs. 22.9 ± 6.3 mL/min/kg, *p* = 0.041), increased dyspnea (NYHA II–III: 69.2% vs. 33.1%, *p* = 0.011), and elevated NT-proBNP levels (median 538 ng/L vs. 215 ng/L, *p* = 0.028);No significant association was found between anemia and demographic characteristics such as age (*p* = 0.389), sex (*p* = 0.668) or underlying congenital heart defect subtype (*p* = 0.955).

Patient collective

The sex distribution in our cohort (51.6% male) reflects patterns seen in other large ACHD registries. For example, the CONgenital CORvitia registry (CONCOR) in the Netherlands reported a 50.2% male distribution [14], while similar proportions were noted by Dimopoulos et al. (49.6% male) and Rodríguez-Hernández et al. (56% male) [2,3].

The mean patient age was 33 years, consistent with prior ACHD studies, where median ages typically ranged between 30 and 36.5 years. In contrast, heart failure cohorts from Jankowska et al. and van Haehling et al. featured older populations with mean ages of 55 and 69 years, respectively, and marked male predominance (88% and 75%) [4].

By exploring the relationship between anemia and functional capacity across ACHD patients, we aimed to identify clinically relevant correlations that may guide risk stratification and management strategies. These findings highlight an under-recognized but potentially modifiable comorbidity in this growing patient population.

Our results emphasize the importance of further prospective studies to better understand the pathophysiology, prognostic implications, and therapeutic potential of anemia and iron deficiency in ACHD. Addressing this knowledge gap may ultimately contribute to improved outcomes and personalized care for adults living with congenital heart disease.

Iron deficiency

According to the definition of iron deficiency (ID) in heart failure proposed by Anker et al. [1], 34% of male and 85% of female ACHD patients in our cohort met the criteria for ID. In contrast, the definition used by Peyrin-Biroulet et al. in their review on ACHD [13] differentiates between functional and absolute iron deficiency, yielding a different distribution: 17% of males and 43% of females were classified as iron deficient, corresponding to an overall prevalence of 29%.

Regardless of the diagnostic criteria used, a significantly higher proportion of female patients were affected by iron deficiency (*p* < 0.001). While this may reflect menstrual blood loss and increased requirements, the broader implications of ID remain uncertain in ACHD. Despite its high prevalence, ID was not associated with reduced exercise capacity or NYHA class in our study, contrasting with consistent findings in chronic heart failure populations [9,14]. One explanation may be that standard ID thresholds lack diagnostic accuracy in ACHD. In patients with cyanosis or erythrocytosis, ferritin can be elevated by inflammation and TSAT suppressed by hypertransferrinemia [1,7,10]. These limitations highlight the need for ACHD-specific markers or thresholds to improve risk stratification and therapeutic targeting.

While anemia was relatively uncommon in our cohort, it demonstrated consistent associations with functional impairment and elevated NT-proBNP. This supports the biological plausibility of hemoglobin as a determinant of oxygen delivery and peak aerobic capacity. The prognostic impact of anemia in ACHD has been underscored in multiple cohorts [19,20] and supports systematic monitoring even in mildly symptomatic patients.

As expected, ID was associated with impaired erythropoiesis in our cohort. Patients with ID had significantly lower hemoglobin concentrations (*p* < 0.001) and reduced mean corpuscular volume (MCV) (*p* < 0.001), consistent with impaired iron availability. In addition, the prevalence of anemia was significantly higher among patients with ID than in those without (*p* = 0.014).

According to the definition of iron deficiency proposed by Anker et al. [1], we did not observe a significant association between iron deficiency and NYHA functional class in our ACHD cohort (*p* = 0.390). This contrasts with earlier findings in chronic heart failure populations, where iron deficiency has been shown to correlate with symptom severity. The discrepancy may reflect differences in age, underlying pathophysiology, or disease perception in ACHD patients compared to older heart failure populations. No significant correlation was found between the type of congenital heart defect and the presence of iron deficiency, reinforcing the notion that ID is influenced more by systemic and demographic factors than by anatomical subtype.

To differentiate our findings from the study by Baumgartner et al. [1], several key distinctions should be emphasized. While their cohort comprised 538 ACHD patients, detailed information regarding age distribution and clinical complexity was not available. Therefore, direct comparisons are limited. Nonetheless, our focused assessment of anemia and iron deficiency may offer complementary insights into this subgroup. Importantly, our study specifically focuses on the interplay between anemia, iron deficiency, and objectively measured exercise capacity, offering granular insight into how hematologic disturbances affect functional outcomes in ACHD. This targeted investigation sets our study apart and provides a novel contribution to the field, beyond the broader scope of previous registry-based analyses.

Prevalence Comparisons


**Anemia in Adults with Congenital Heart Disease:**


Anemia was present in 13 patients (4.2%) of ACHD patients (9 female (6.0%) and 4 male (2.5%), with microcytic morphology in 46% and normocytic in 54% of cases. Approximately two-thirds of anemic patients also had iron deficiency, suggesting ID as the predominant cause. A strong association between low MCV and anemia (*p* < 0.001) supports this link and aligns with findings from Rodríguez-Hernández et al. and Dimopoulos et al. [1,2].

Anemic patients were more likely to exhibit NYHA functional class > I (*p* = 0.045), elevated NT-proBNP levels (651 ± 831 ng/L vs. 180 ± 353 ng/L, *p* = 0.010), reduced transferrin saturation (TSAT) (*p* = 0.003), and lower ferritin levels (*p* = 0.030) compared to non-anemic counterparts. Additionally, they had higher usage of ACE inhibitors, AT1 antagonists, and beta-blockers, suggesting that patients with greater clinical severity are more susceptible to anemia. Conversely, anemia itself may contribute to symptom burden, as evidenced by its association with dyspnea and reduced exercise capacity in this cohort.

Prevalence Comparisons


**Functional Capacity in Adults with Congenital Heart Disease (ACHD):**


While studies in chronic heart failure (HF) populations have demonstrated a significant association between iron deficiency (ID) and maximal oxygen consumption (VO_2_ max) [1], no such correlation was observed in our ACHD cohort (*p* = 0.741). However, a modest but significant association was identified between serum ferritin levels and METs achieved during ergometry (r = 0.201, *p* = 0.048).

While no association with exercise metrics such as max VO_2_ (*p* = 0.058) or METs in ergometry (*p* = 0.110) was initially found, a statistically significant relationship between anemia and max VO_2_ emerged after the exclusion of Eisenmenger patients.

A more robust and clinically relevant correlation was observed between hemoglobin (Hb) concentration and exercise performance. Hb levels showed a significant positive correlation with both VO_2_ max (r = 0.221, *p* = 0.006) and METs achieved during ergometry (r = 0.292, *p* = 0.004), suggesting that even subclinical reductions in hemoglobin may impact functional capacity in ACHD patients. These results underline the physiological importance of hemoglobin in determining aerobic performance and support its potential role as a simple but powerful marker in clinical risk assessment.

The higher prevalence of anemia in female patients may be attributable to physiological factors such as menstrual blood loss. Given the relatively young age of our cohort, this explanation is plausible. Epidemiological data report anemia prevalence in reproductive-age women between 6% and 20%, depending on the region and criteria used [12]. Our findings (8.0%) fall within this expected range and underscore the need for sex-specific screening considerations in ACHD.

As expected, NYHA functional class was strongly associated with exercise capacity. Patients with NYHA class > I had significantly reduced performance across all modalities (*p* < 0.001), with an average VO_2_ max of 16.7 ± 4.5 mL/min·kg and METs of 4.5 ± 0.6, compared to 24.2 ± 6.0 mL/min·kg and 8.0 ± 2.0, respectively, in NYHA class I patients.

Finally, NT-proBNP levels correlated negatively with exercise capacity, confirming its value as a functional biomarker. This association was significant for both, as follows:Spiroergometry VO_2_ max: r = −0.507, *p* < 0.001;Ergometry METs: r = −0.341, *p* = 0.001.

Comparison of Prevalence

Although routine iron status screening and intravenous iron supplementation are recommended in chronic HF management, our findings do not support direct extrapolation of these guidelines to the ACHD population. Instead, hemoglobin concentration and the presence of anemia—rather than isolated iron deficiency—may serve as more reliable indicators of impaired exercise capacity in ACHD.

These data highlight the importance of individualized management strategies in ACHD and caution against the blanket application of heart failure treatment protocols to this distinct population.

Limitations

Moreover, the single-center nature of this study may reduce the generalizability of our findings. However, such a design is common in ACHD research given the rarity of these patients and logistical challenges in multi-institutional data harmonization [7]. Our cohort is well-characterized and includes detailed iron status and CPET parameters that are infrequently available in routine clinical datasets. Although anemia was infrequent, the prevalence aligns with previous reports [19,20]. These findings should be validated in future multicenter prospective registries with uniform definitions of ID tailored to ACHD [10].

Moreover, this was a single-center study, which may affect generalizability. Future prospective, multicenter studies involving larger cohorts are needed to validate these findings and ensure broader applicability to the diverse ACHD population.

## 5. Conclusions

The prevalence of iron deficiency (59.0%) and anemia (4.2%) in adults with congenital heart disease (ACHD) was in line with prior reports. While no significant association was observed between ID and exercise capacity, reduced hemoglobin levels—particularly in anemic patients—were consistently associated with impaired functional performance when patients with Eisenmenger syndrome were excluded. In contrast, lower hemoglobin levels were significantly associated with impaired functional performance, suggesting that hemoglobin may be a more reliable marker of reduced exercise capacity than ID alone in this population.

These findings highlight the importance of the systematic evaluation of iron status and hemoglobin concentration in ACHD patients, particularly in those reporting functional limitations. They also underscore the need for prospective studies to investigate whether iron supplementation or anemia-targeted therapies could improve exercise tolerance and clinical outcomes in this growing patient group.

## Figures and Tables

**Figure 1 diagnostics-15-01672-f001:**
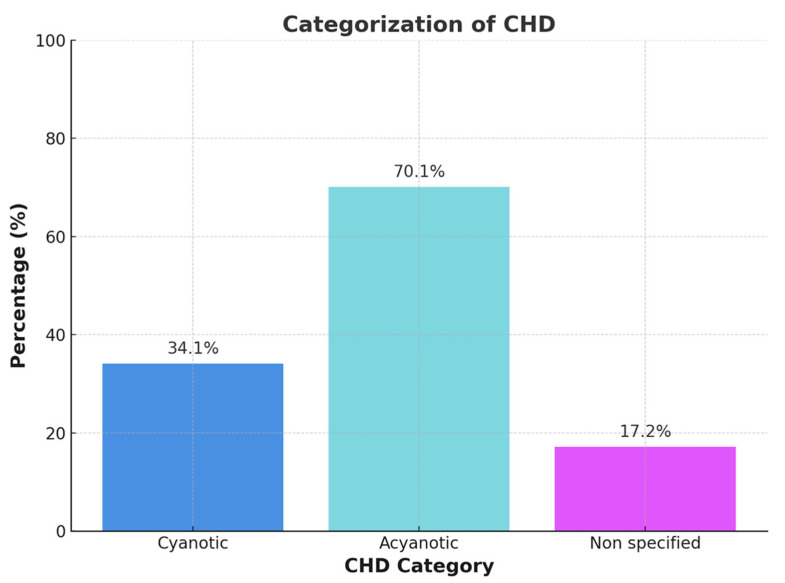
Categorization of congenital heart disease into cyanotic, acyanotic, und non-specified heart disease.

**Figure 2 diagnostics-15-01672-f002:**
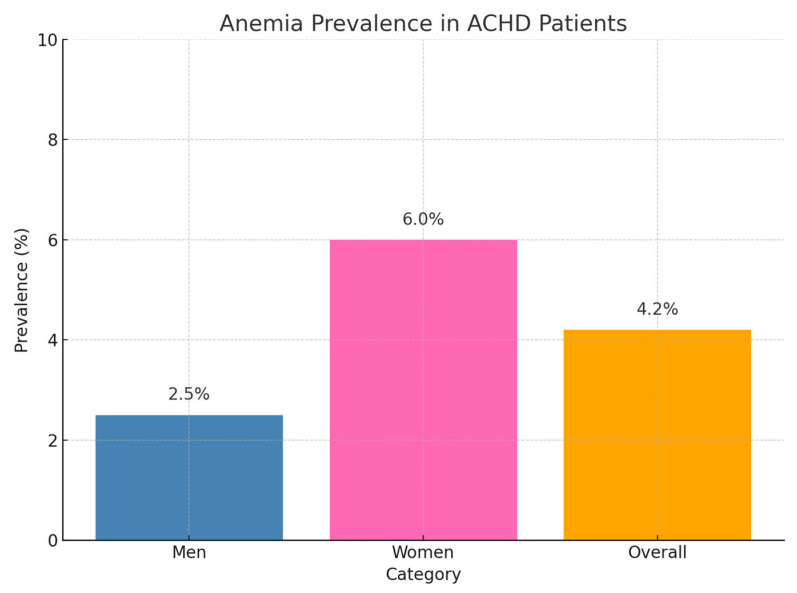
Anemia in ACHD.

**Figure 3 diagnostics-15-01672-f003:**
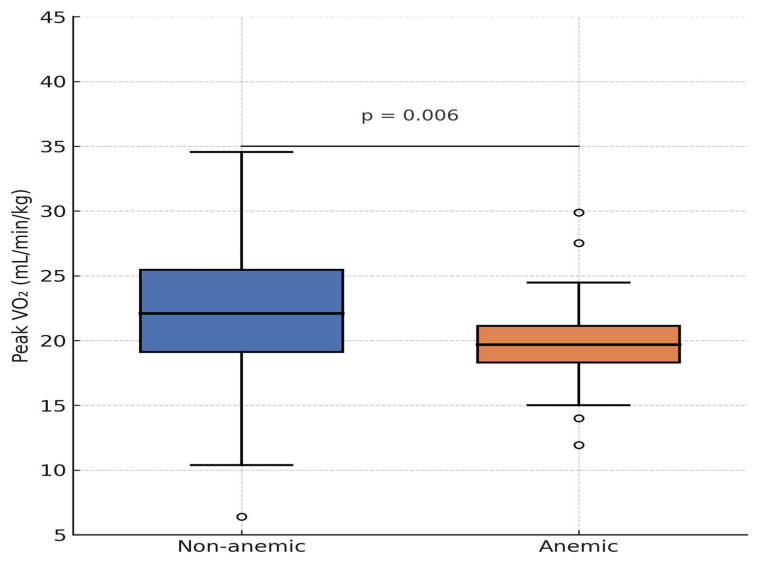
Distribution of maximal oxygen uptake per body weight depending on anemia.

**Figure 4 diagnostics-15-01672-f004:**
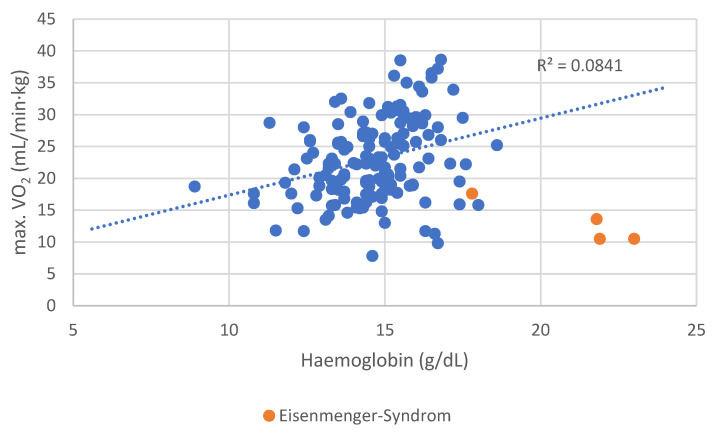
Distribution of maximal oxygen uptake per body weight depending on the hemoglobin level.

**Figure 5 diagnostics-15-01672-f005:**
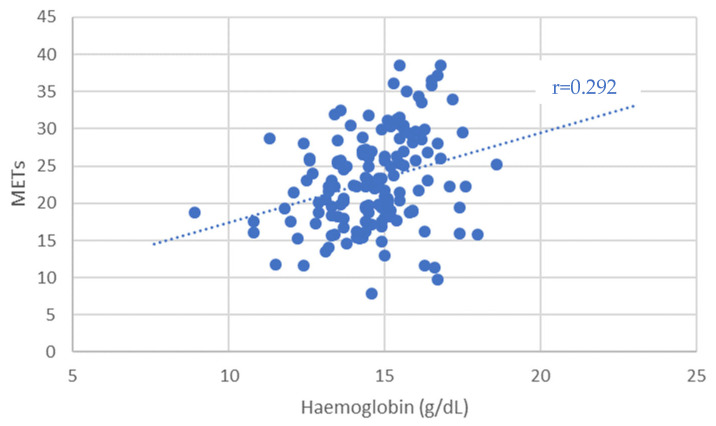
Distribution of METs depending on the hemoglobin level.

**Table 1 diagnostics-15-01672-t001:** Baseline demographics and clinical characteristics.

Baseline Characteristics	Men		Women		Total	
	$\bar{x}$ ± σ	N	$\bar{x}$ ± σ	N	$\bar{x}$ ± σ	N
Age, y	33 ± 11	160	33 ± 12	150	33 ± 12	310
Height (cm)	177 ± 8	159	165 ± 9	142	171 ± 11	301
Weight (kg)	81 ± 16	160	66 ± 14	140	74 ± 17	300
BMI (kg/m^2^)	25.8 ± 4.6	159	24.4 ± 5.0	141	25.1 ± 4.9	300
Shunt lesions	-	53	-	53	-	106
Left-Sided Malformations	-	63	-	58	-	121
Right-sided Malformations	-	66	-	88	-	154
Complex Malformations	-	81	-	63	-	144

**Table 2 diagnostics-15-01672-t002:** Distribution of congenital heart defects according to ESC categories (N = 310).

ESC Category	Diagnosis	Abbreviation	%	N
Shunt Lesions	Ventricular Septal Defect	VSD	10.6	33
	Atrial Septal Defect	ASD	10.3	32
	Persistent Ductus Arteriosus	PDA	7.4	23
	Atrioventricular Septal Defect	AVSD	7.4	23
	Anomalous Pulmonary Venous Connection	APVC	4.5	14
	Subtotal		40.2	125
Left-sided Malformations	Aortic Isthmus Stenosis	CoA	14.8	46
	Aortic Valve Stenosis	AS	6.1	19
	Supravalvular Aortic Stenosis	Suprav. AS	0.3	1
	Congenital Mitral Valve Defects	-	2.3	7
	Aortopathy	-	4.5	14
	Subtotal		28.0	87
Right-sided Malformations	Pulmonary Valve Stenosis	PS	11.3	35
	Ebstein Anomaly	-	2.6	8
	Double-Chambered Right Ventricle	DCRV	1.0	3
	Coronary Anomalies	-	1.3	4
	Subtotal		16.2	50
Complex Malformations	Tetralogy of Fallot	TOF	15.8	49
	Transposition of the Great Arteries	TGA	10.0	31
	Congenitally Corrected TGA	ccTGA	1.9	6
	Double Outlet Right Ventricle	DORV	2.6	8
	Truncus Arteriosus	TAC	1.6	5
	Hypoplastic Left Heart Syndrome	HLHS	0.3	1
	Pulmonary Atresia ± VSD	PA ± VSD	1.6	5
	Tricuspid Atresia	TrA	1.9	6
	Double Inlet Ventricle	DIV	0.3	1
	Subtotal		36.0	112

**Table 3 diagnostics-15-01672-t003:** Comorbidities of patients.

Comorbidity	Percentage (%)	Total	SL (%)	LF (%)	RF (%)	CF (%)	*p*-Value
Arterial Hypertension	18.1	56	14.7	28.6	15.5	13.5	0.334891
Pulmonary Hypertension	5.2	16	6.7	-	2.4	12.2	0.010101
Chronic Kidney Insufficiency	1.6	5	1.3	1.3	2.4	2.7	1.000000
Diabetes Mellitus	1.3	4	4.0	-	-	1.4	0.333333
Trisomy 21	4.8	15	10.7	-	3.6	5.4	0.060606
Bronchial Asthma	5.5	17	4.0	9.1	3.6	5.4	0.151515
Chronic Obstructive Pulmonary Disease	1.0	3	2.7	-	-	1.4	1.000000
Migraine	6.1	19	6.7	11.7	2.4	4.1	0.242424
Chronic Inflammatory Bowel Disease	0.3	1	-	-	1.2	-	1.000000
Celiac Disease	0.3	1	-	1.3	-	-	1.000000
Gastritis, Esophagitis, Enteritis	0.6	2	-	1.3	-	1.4	1.000000
Reflux	0.3	1	-	1.3	-	-	1.000000
Polymenorrhea	1.0	3	-	-	3.6	-	1.000000
Depression	4.8	15	12.0	2.6	1.2	4.1	0.151515

**Table 4 diagnostics-15-01672-t004:** Medication for patients.

Medication	Percentage (%)	Total	SL (%)	LF (%)	RF (%)	CF (%)	*p*-Value
Anticoagulant	10.6	33	12.0	3.9	4.8	23.0	0.102
Antiplatelet Agent	6.1	19	2.7	7.8	7.1	6.8	0.632
Iron Supplement	1.6	5	1.3	-	1.2	4.1	0.825
ACE Inhibitor	11.0	34	10.7	13.0	3.6	17.6	0.467
AT1 Antagonist	7.7	24	5.3	9.1	9.5	6.8	0.597
Calcium Channel Blocker	2.9	9	5.3	3.9	1.2	1.4	0.439
Beta Blocker	19.4	60	16.0	16.9	19.0	25.7	0.901
Loop Diuretic	5.2	16	4.0	-	6.0	10.8	0.762
Thiazid Diuretic	6.1	19	4.0	7.8	4.8	8.1	0.931
Potassium-Sparing Diuretics	2.9	9	1.3	-	2.4	8.1	0.448
Proton Pump Inhibitor	3.5	11	4.0	5.2	3.6	1.4	0.483
Oral Contraceptive	6.5	20	5.3	5.2	4.8	10.8	0.954
NSAID	1.3	4	1.3	-	3.6	-	0.852

**Table 5 diagnostics-15-01672-t005:** Laboratory values of the patient collective.

	Men		Women		Total	
	$\bar{x}$ ± σ	N	$\bar{x}$ ± σ	N	$\bar{x}$ ± σ	N
Leukocytes (Thous/µL)	6.89 ± 1.89	160	7.25 ± 2.03	150	7.06 ± 1.96	310
Erythrocytes (Mill/µL)	5.26 ± 0.50	159	4.70 ± 0.51	150	4.98 ± 0.58	309
Hemoglobin (g/dL)	15.7 ± 1.5	160	13.8 ± 1.7	150	14.8 ± 1.9	310
Hematocrit (%)	46.1 ± 4.8	160	41.4 ± 4.6	150	43.8 ± 5.3	310
MCV (fL)	87.5 ± 4.7	160	88.2 ± 5.7	150	87.8 ± 5.2	310
MCHC (g/dL)	34.1 ± 1.1	160	33.2 ± 1.2	150	33.7 ± 1.2	310
Platelets (Thous/µL)	224 ± 50	160	255 ± 67	150	239 ± 60	310
Ferritin (µg/L)	209 ± 279	160	58 ± 51	150	136 ± 216	310
TSAT (%)	29.7 ± 11.6	160	23.2 ± 11.6	150	26.6 ± 12.0	310
Transferrin (mg/dL)	249 ± 34	160	288 ± 52	150	268 ± 48	310
sTfR (mg/L)	1.30 ± 0.77	144	1.37 ± 0.62	144	1.34 ± 0.70	288
Iron (µg/dL)	102 ± 35	160	91 ± 44	150	97 ± 40	310
Troponin (ng/L)	5.16 ± 6.43	159	2.01 ± 5.33	150	3.63 ± 6.12	309
NTproBNP (ng/L)	146 ± 214	160	256 ± 516	150	200 ± 394	310
Bilirubin (mg/dL)	0.72 ± 0.47	156	0.56 ± 0.41	146	0.64 ± 0.45	302
CRP (mg/dL)	0.25 ± 0.76	159	0.30 ± 0.59	150	0.27 ± 0.68	309
LDH (U/L)	206 ± 52	160	204 ± 62	150	205 ± 57	310
Creatinine (mg/dL)	0.98 ± 0.16	160	0.78 ± 0.17	150	0.88 ± 0.19	310
eGFR (mL/min)	102 ± 16	158	102 ± 19	150	102 ± 17	308
TSH (mU/L)	2.47 ± 2.48	158	1.99 ± 1.15	149	2.24 ± 1.96	307
INR	1.14 ± 0.36	159	1.21 ± 0.49	150	1.17 ± 0.43	309

## Data Availability

All data are included in the manuscript.

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
