# Peer review of "Iron Status, Anemia, and Functional Capacity in Adults with Congenital Heart Disease"

_diagnostics, 2025, doi:10.3390/diagnostics15131672_

Round 1
Reviewer 1 Report
Comments and Suggestions for Authors
This retrospective analysis of iron deficiency and anemia’s influence on functional status in adults with congenital heart disease, while substantial from the clinical perspective, still suffers from significant methodological and interpretative limitations. The authors indeed find the strikingly high prevalence of iron deficiency and the rather low rate of anemia, noting a robust correlation of the latter to the impairment of exercise capacity assessed as VO 2 max. The correlation with iron deficiency appears much weaker. The paper, based on data collected for 310 adults with congenital heart disease, offering stratified analysis for the cyanotic and acyanotic subtypes, has solid figure background yet suffers dramatically from the limitation to the single-center retrospective design, potentially decreasing the generalizability of its findings. Moreover, the major of the statistical findings, such as the impairment of VO 2 max as related to anemia, are statistically robust yet, in clinical terms, do not offer significant added value due to extremely low number of the anemic subgroup; the studies offer a matching subgroup of iron deficiency patients. Notably, the authors utilize the established definitions of iron deficiency from the heart failure literature, which may be hardly applicable to the congenital heard disease population, particularly to such confounders as chronic hypoxia or erythrocytosis seen in Eisenmenger syndrome patients, which are excluded from the analysis. However, the narrative often jumps in
Reviewer 2 Report
Comments and Suggestions for Authors
The topic of the manuscript is crucial because the authors aimed to assess the prevalence of iron deficiency and anemia in patients with congenital heart disease and examine their association with objective measures of functional capacity. Anemia and iron deficiency are known risk factors of the low level of physical capacity and worsen prognosis, especially among patients with heart failure, but their role in adults with congenital heart diseases are not well studied. The authors presented retrospective analysis of 310 outpatients with different types of congenital abnormalities and detailed their clinical and laboratory status. It was revealed, that the prevalence of iron deficiency was 59.0%, and anemia - 4.2% (8,0% of females, 2,5% of males) in adults with congenital heart disease, and iron deficiency was the main cause of anemia. The authors concluded, that low concentration of hemoglobin can be as one of reliable marker of reduced exercise capacity compared with iron deficiency alone.
There are several minor comments:
1. Could you be so kind to add the reference in this sentence (Systolic ventricular function was categorized based on ejection fraction (EF): normal 109 (≥55%), mildly reduced (45–54%), moderately reduced (30–44%), and severely reduced 110 (<30%).
2. It's not completely clear, why the quantity of patients is 310, but not is 296? (due to text: 525 patient visits fulfilled the study inclusion criteria. Of these, 215 were excluded as repeat follow-up visits from patients already represented in the dataset, ensuring that only one visit per patient was included in the analysis. An additional 14 cases were excluded due to missing iron panel data (ferritin and transferrin saturation), resulting in a final cohort of 310 unique patients with complete iron status and exercise capacity data).
3. The list of references should be updated - 47,4% of references were published more than 10 years ago.
Reviewer 3 Report
Comments and Suggestions for Authors
Dear Authors,
The manuscript entitled "Iron Status, Anemia, and Functional Capacity in Adults with Congenital Heart Disease"addresses a highly relevant and underexplored aspect of ACHD management. The topic is of great clinical importance and certainly deserves further investigation. However, in its current form, the paper requires significant revision to improve clarity and accuracy.
The methodology section lacks sufficient references. Each clinical parameter and diagnostic criterion used should be supported by appropriate literature. For example, the assessment of ejection fraction and the diagnosis of heart failure should be based on current ESC guidelines. Please ensure that all evaluations and classifications are properly referenced.
Every parameter measured or discussed, such as iron status indicators, hemoglobin levels, and functional capacity assessments, should be accompanied by a reference that justifies its clinical relevance and threshold values.
The results section is rather scarce and would benefit from further revision. In some instances you use odd numbers. while in oder you use percentages. Please make it uniform.
Moreover, you state that there are 13 anemic patients; however, the breakdown indicates 4 males and 12 females, which totals 16 patients. This discrepancy should be clarified.
Since a larger proportion of anemic patients are female, it would be valuable to discuss whether this could be attributed to physiological factors, such as regular menstrual cycles. A comparison with anemia prevalence in the general population would provide further context and strengthen your discussion.
Sincerely
Round 2
Reviewer 3 Report
Comments and Suggestions for Authors
The revision has significantly improved the quality and clarity of the manuscript. In its current form, the paper is acceptable for publication.